# Rationale and study protocol of a regional health panel in Saxony, Germany (GEPASA)

**Lorenz Harst** ⓘ *◎, **Tina Haase** ⓘ ◎, **Falko Tesch, Lilly Rüthrich, Markus Kösters**‡, **Jochen Schmitt**‡

Center for Evidence-Based Healthcare, University Clinic and Faculty of Medicine Carl Gustav Carus, Technische Universität Dresden, Dresden, Germany

◎ These authors contributed equally to this work.
‡ MK and JS also contributed equally to this work.
* Lorenz.Harst@ukdd.de

**Data Availability Statement:** No datasets were generated or analysed during the current study. All relevant data from this study will be made available upon request after study completion.

## Abstract

### Background

The citizens' perspectives on health care are central to the assessment of the health care situation and to regional development. In Germany, however, strategic goals for health care delivery are planned based on population statistics and partly on regional morbidity. Saxony is a German federal state with high average age and low density of physicians which makes the population perspective on quality of health care especially intersting. No existing panel surveys cover issues related to the perceived quality of health care delivery on a regional level in Germany.

### Aim

We aim to conduct a longitudinal panel study of the perceived health status and perceived quality of health care of the Saxon citizens as a basis for the systematic derivation of health care goals/measures and target group-specific, regionally suitable prevention measures.

### Methods

With an anticipated 15% response rate, 15,000 potential participants have to be contacted to achieve a calculated sample size of about 2,250 participants. The questionnaire will be circulated every other year with the option for adaptations depending on insights of each panel wave. The study protocol was approved by the local ethics committee of the Technische Universität Dresden, Germany (ethical approval code BO-EK-320072022) and has been registered at the German Clinical Trials Register as the German WHO primary registry (study registration number DRKS00031229).

### Discussion and outlook

The results are intended to identify gaps in health care and to develop patient-centered health care goals for the region together with stakeholders in Saxon health care planning. At the same time, longitudinal data allow mapping of perceived health status and perceived health care trends.

**Funding:** The author(s) received no specific funding for this work.

**Competing interests:** I have read the journal's policy and the authors of this manuscript have the following competing interests: JS reports institutional grants for investigator-initiated research from the German GBA, the BMG, BMBF, EU, Federal State of Saxony, Novartis, Sanofi, ALK, and Pfizer. He also participated in advisory board meetings as a paid consultant for Sanofi, Lilly, and ALK. JS serves the German Ministry of Health as a member of the Sachverständigenrat Gesundheit und Pflege. MK reports institutional grants for investigator-initiated research from the German GBA, the EU, BMBF and the DFG. LH reports fees from the Volkswagen Foundation and from Thieme. LR and TH report fees from the aQua-Institut. These reports are unrelated to this study. This does not alter our adherence to PLOS ONE policies on sharing data and materials.

# 1. Introduction

Perceived attractiveness of a region is determined by how citizens rate their quality of life [1]. Increasing quality of life and attractiveness of a city or region will increase the number of people moving into the area while decreasing the number of people moving out, resulting in the settlement of businesses and cultural facilities [2]. Health status and continuously ensured local health care are prerequisites for a perceived high quality of life [1]. Understanding the citizens' perspectives on health care delivery is central to the assessment of the health care situation and to regional development throughout the world [3], and the population's perspective should be taken into account when developing strategic goals for health care delivery [4]. Considering the citizens' perspectives on health care delivery is also in line with the principles of evidence-based health care and value-based health care. Both concepts stress the importance of patient-centeredness in the assessment of health care structures as well as in the development of health care innovations and when it comes to a quality-orientated renumeration of health care services [5, 6]. A basic prerequisite is the measurement of quality using quality indicators, which necessarily have to include a patient perspective, ideally measured using patient-reported experience measures [7]. In the U.S., reimbursement of health care services based on such quality indicators is already standard practice in many areas and ensures that only high-quality health care services are reimbursed [8]. Taking a wider perspective, patient-centeredness in health care planning is a necessary prerequisite for patient empowerment as one feature of modern health care delivery [9].

In Germany's federal health care system, however, health care delivery is currently planned solely based on demographics as well as morbidity of certain chronic diseases in a given federal state (§9 section 2 BPL-RL [10]). Although a recent expert statement on the development of health care planning in Germany suggests to consider wait times and travel distances in health care planning [11], the population's perspective on the availability of doctors, hospitals or other health care facilities, on wait times or on travel distances to health care facilities is still not considered within the federal planning process. These issues, however, are dimensions of quality of health care, covering structural (e.g. availability, distances), process (e.g. waiting times) as well as outcome (experiences made during consultations with health care professionals) quality alike [12]. While some of these domains (availability, distances) can be measured using, e.g., routine data from health care, others can only be assessed by asking (potential) patients directly about them [13].

## 1.1 Rationale

Saxony is one of the German federal states with the highest average age. Of its four million people population, 27% are older than 65 years (reference 2021) and Saxony has the 6th lowest birthrate of all German federal states (1.527 in 2021) [14]. The proportion of the age group older than 65 years is lowest in and around the cities of Leipzig and Dresden and highest in more rural areas, sometimes reaching over 30% (e.g. in the districts of Görlitz, Zwickau, Vogtland, Central Saxony). At least partly, the latter are also the regions where density of physicians and probability of settling new physicians is lowest [15]. This is of particular importance as the proportion of doctors reaching retirement age in the near vicinity (those who are already over 59 years old) varies between 50% and 75% throughout Saxony [15]. The expectedly decreasing number of physicians meets a growing demand for health care. An increase in the number of patients with diabetes mellitus, hypertension, dementia or oncological diseases is predicted for almost all regions of Saxony [15]. Additionally, Saxony had the 3rd largest death rate of all German federal states, independent of the cause of death (1,480 deaths per 100,000 citizens in 2022) [14].

Panel surveys are longitudinal studies researching the same sample time and again over a potentially open-ended time period. By collecting longitudinal data on the topics raised above, panel surveys allow i) for mapping changes in the assessment of various outcomes at aggregate and individual data levels [16], ii) for avoiding measurement bias due to single points of measurement [17] and iii) for conducting causal analyses on the effectiveness of health care interventions (if intervention and control groups are formed) (see e.g. [18]). Hence, panel surveys contribute to the development of political agendas and their alignment with the reality of citizens' lives throughout Europe. For example, data from the European Social Survey, which explicitly covers health and well-being every two years, are widely used to assess social determinants of health, which is a prerequisite for targeted social policies [19]. The Health Survey for England, conducted regularly by the NHS, allows e.g., for determining predictors of health care access as well as health outcomes [20]. However, none of the panel questionnaires mentioned above cover issues releated to the perceived quality of health care delivery within a region.

### 1.2 Objective

Therefore, we aim to conduct a structured and longitudinal survey of perceived health status and perceived quality of health care of citizens in Saxony as the first health care panel for Saxony. The panel data can serve as a basis for the systematic derivation of health care goals and target group-specific, regionally appropriate preventive measures and interventions to maintain or improve the quality of care in the region.

The panel's overarching research questions are:

1. What is the relationship between perceptions of health status, quality of health care and quality of life?

2. Do individual (health) behavior, use of health care facilities, medical diagnoses, work environment and socio-demographic factors influence perceptions of health status, quality of health care and quality of life?

3. How do perceptions of health status, quality of health care and quality of life change over time?

A general overview of the domains and research questions is depicted in Fig 1.

Apart from answering the above listed research questions, our procedure is aimed to be explorative, aiming to shine a light on regional perceptions of quality of life and its predictors and to develop hypotheses to be tested in futher panel waves.

## 2. Methodology

In order to answer the above-mentioned research questions and conduct longitudinal assessments of perceived health status, quality of health care and quality of life in Saxony, we plan to set up a regional health panel in Saxony. The reporting of our methodology follows the PLOSOne_Clinical_Studies_Checklist (see checklist in supporting file S1 File "PLOS ONE Clinical Studies Checklist").

### 2.1 Domains covered in the questionnaire

The general questionnaire to be used in each wave of the panel will cover the following issues:

- perceived quality of life at the region of residence (dependent variable used for studying the influence of perceived quality of health care on quality of life at the region of residence)

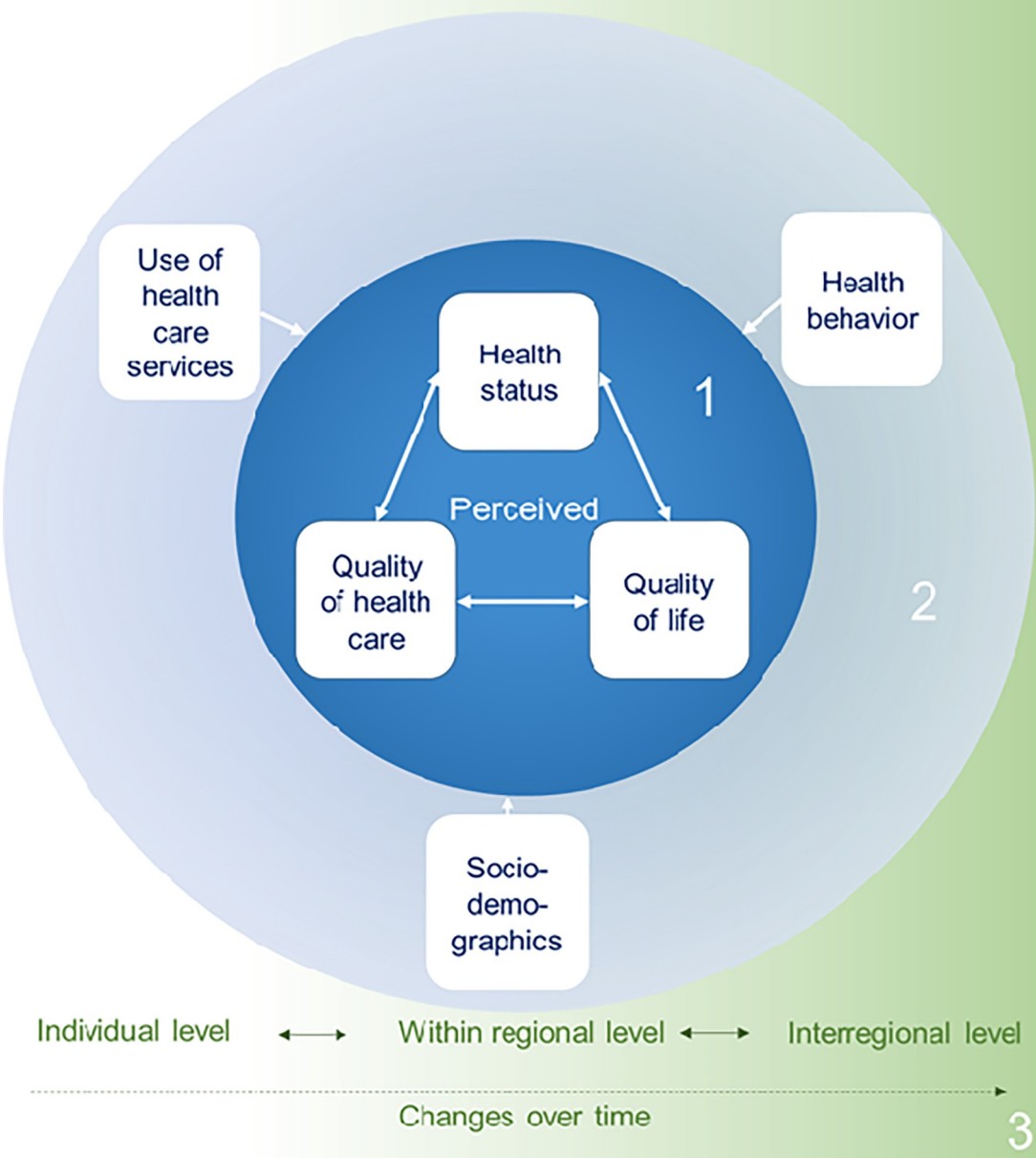

**Fig 1. Domains and research questions.** The panel's research questions are 1) What is the relationship between perceptions of health status, quality of health care and quality of life? 2) Do individual (health) behavior, use of health care facilities, medical diagnoses, work environment and socio-demographic factors influence perceptions of health status, quality of health care and quality of life? 3) How do perceptions of health status, quality of health care and quality of life change over time?

- perceived health status and health problems (perceived health problems and specific diagnoses; predictor for utilization of health care services and perceived quality of health care)

- utilization of health care services (predictor for perceived quality of health care)

- perceived quality of health care (predictor for quality of life at the region of residence and central construct of interest of the panel survey)

- behavioral habits (predictor for perceived health status and utilization of health care services)

- education, occupation and household status (dimension of living conditions; predictor for utilization of health care services and perceived health status)

- demographics (insurance and relationship status, gender, age, weight, height; predictor for utilization of health care services and perceived health status)

The complete questionnaire, containing both the questions (n = 48) and the scales, is available as a supplementary file to this publication (see supporting file S2 File "Complete Questionnaire"). Within this file, we also report on the origin of each question, thus demonstrating we only use standardized–not open-ended–questions and mostly rely on previously used and such valid as well as reliable questions. We also explain each question's anticipated use in subsequent data analyses. Thus, we also reflect on the role each question plays in the overall purpose of our panel study.

## 2.2 Sample size planning

Due to the explorative character of our work, no definitive hypotheses and therefore endpoints can be predefined. Therefore, sample size planning was done based on an exemplary correlation: The sample size was planned to be able to detect an effect of OR = 1.50 with a power of 90% within a regression analysis for a binomially distributed endpoint with a prevalence of 20% and an alpha error of 5% (two-sided). This assumption is based on the example of the Global Activity Limitation Indicator (GALI), used by the EU to measure Healthy Life Years [21], assuming about 20% of the Saxonian population with activity limitations. Sample size was calculated so that this proportion would be reprensented within the sample.

This results in a required sample size of n = 2,196. Assuming a 15% response rate about 14,640 participants (rounded to 15,000) subjects have to be recruited in the first wave.

## 2.3 Sampling

A single stage stratified sample will be drawn for the panel. A selection of 16 medium-sized cities in Saxony with at least 15,000 inhabitants in the year 2020 will be represented in the sample. These are expected to experience a particularly sharp decline in available medical care providers, while at the same time forming the spatial planning units for health care provision in Saxony. As a result, rural communities will be slightly underrepresented in the sample.

In addition to 16 medium-sized cities, 16 randomly selected districts of the four large cities Dresden, Leipzig, Chemnitz and Zwickau as well as 16 randomly selected municipalities will be drawn into the sample. Municipalities largely represent rural areas in Saxony and were drawn based on their postal code. In order to ensure a clear differentiation of the area of residence of the participants, municipalities with identical postal codes were not included in the sample. Thus, the final sample is made up from 48 clusters. For each of the 48 clusters formed, 312 to 313 persons will be randomly selected for initial contact. For this purpose, a random sample is requested from the relevant residents' registration offices for each district or municipality. Administrative fees for drawing the sample are anticipated for some of the registration offices.

Changes in coverage in a particular municipality or district can thus be compared with other municipalities/districts with a similarly large sample. Design effects are estimated to be negligible because the sample clusters cover a fairly large geographic unit.

The sample construction with reference to the Saxon population can be found in Table 1.

## 2.4 Recruitment and study conduct

Initially, selected individuals will be contacted by a postal cover letter including general information on the panel, the data protection and privacy concept, the questionnaire for the first

**Table 1. Sample construction with reference to the Saxon population.**

| Region | Cluster | Sample | Residents 2020 | Portion of region | Portion of sample |
|---|---|---|---|---|---|
| Leipzig | 6 | 1,875 | 592,796 | 14,6% | 12.5% |
| Dresden | 5 | 1,562.5 | 555,459 | 13,7% | 10.4% |
| Chemnitz | 3 | 937.5 | 245,166 | 6,0% | 6.3% |
| Zwickau | 2 | 625 | 88,091 | 2,2% | 4.2% |
| Large cities | 16 | 5,000 | 1,481.512 | 36,5% | 33,3% |
| Medium-sized cities | 16 | 5,000 | 950,893 | 23,4% | 33,3% |
| Municipalities | 16 | 5,000 | 1,630,995 | 40,1% | 33,3% |
| Sum | 48 | 15,000 | 4,063,400 | 100% | 100% |

survey wave as well as a consent form. Participants are therein requested to give informed consent for participation in the panel for its entire duration. Letters to the individual participants are addressed and sent by the Data Trust Office of Technische Universität (TU) Dresden (see below) in order to ensure separation of the participants' personal information and their responses in the questionnaire. Only those participants sending back a signed consent form will subsequently be included in the panel. All return letters are collected at the project office within the responsible institute, where questionnaire and consent forms are separated. The latter are collected and managed by the Data Trust Office.

The questionnaire will be presented in paper and will also be available as an online version. The printed questionnaire will be created using the software FormPro and the digital questionnaire will be created using the software REDCap. Participants choosing the latter option will find an individualized link to the online questionnaire within the recruitement papers. A stamped return envelope is included in the initial contact letter for participants deciding to complete the questionnaire on paper. Data entry from printed questionnaires is largely automated using FormPro software. The accuracy of the readings is checked manually.

In case of a response rate lower than 15%, advertisement in official publications of the major's offices in the included communities will be placed, explaining the purpose of the panel and stressing the fact that participation is a means for citizens to be heard on such an important issue as health care planning.

## 2.5 Data security, privacy and ethics

The data security concept was developed by the project team and reviewed by the data security office of the Medical Faculty Carl Gustav Carus at the TU Dresden and the federal data security office of Saxony. According to the concept, the completed questionnaires–if they were filled out on paper–are digitized and stored on secure servers, separately from the person-identifying data. Person-identifying data, such as names and addresses, are only stored at the Data Trust Office of the TU Dresden, allowing the office to contact the participants for each wave. It is not possible to draw any conclusions from the participants' details to their person during data analysis. This is ensured by coding/pseudonymization. Study staff will not have access to the coding list at any time. It is exclusively created and maintained by the independent Data Trust Office of the TU Dresden.

Ethics approval for this panel survey was granted by the ethics committee of the TU Dresden in September 2022 (No BO-EK-320072022). The study protocol has been registered at the German Clinical Trials Register as the German WHO primary registry in March 2023 (study registration number DRKS00031229).

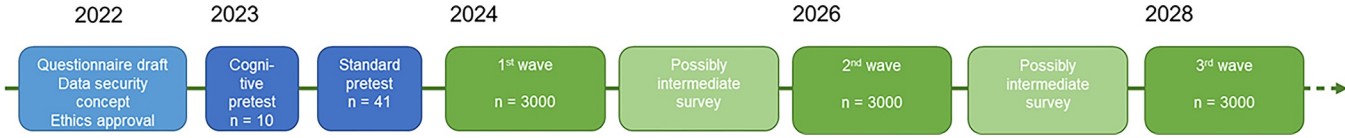

**Fig 2. Timeline.** In 2022, the questionnaire and the data security concept were designed and ethics approval was obtained. Pretests were conducted in 2023. Survey waves are planned every two years, starting in 2024. Intermediate surveys on focus topics are possible.

### 2.6 Status and timeline

Fig 2 depicts the timeline for the panel survey. The survey is designed as an open-ended project.

### 2.7 Statistical plan

Seeing as the panel is not designed to test hypotheses and thus no confirmatory research questions were designed, data analysis will be primarilay descriptive. Additionally, changes over time will be visualized in graphical form. Apart from that, appropriate statistical models will be used to analyze changes in the domains covered by each questionnaire. Thus, Analyses of Variance (ANOVAs) and t-tests will be regularly conducted, should a normal distribution be present. Otherwise, non-parametric alternatives such as the Mann-Whitney-U- or Kruskall-Wallis test will be used. In order to analyze relationships between domains, appropriate correlation measures will be computed with respect to the data available and tests for normal distribution (Cramer's V for nominal scales, Spearman's r or Kendall's tau b for ordinal scales, Pearson's r for metric scales). Where literature suggests causal inferences are likely, Regression models or Strcutural Equation Modelling for latent constructs will be used.

## 3. Discussion

Although calls for citizens' participation in health care planning are unanimous, neither a theoretical basis or a model for participation in health care planning exist so far, nor do standardized sets of methods to achieve participation [22]. The proposed panel project will provide an instrument to systematically research patient perceptions of health care delivery in Saxony and thus provide a solid basis for the development of health care goals on a regional level. Such goals are seen as an important pillar for a sustainable and high-quality health care delivery in Germany [23]. In line with the central requirement that health care goals be measurable, the solid database to be provided by this panel project is also a component of continuous regionalized health care monitoring [24]. Since the same set of questions is to be used in each wave of the panel survey, the proposed research questions can be assessed over time, thus providing a theoretical foundation for citizen participation in health care. Also, by mostly using previous used and such valid and reliable questions, the results of the panel surveys can be compared to at least national benchmarks. Furthermore, data gained on perceived quality of health care can be used complementary to routinely collected data on health care and such contribute to a more holistic health care monitoring [25]. In this way, the results will contribute to the growing body of health care science, which a) aims to build theories for change in health care planning, and b) has patient benefit as its primary directive [26].

## Supporting information

**S1 File. PLOS ONE clinical studies checklist.**
(PDF)

**S2 File. Complete questionnaire.**
(PDF)

## Acknowledgments

The authors would like to thank the Medical Faculty Carl Gustav Carus Dresden, the State Chamber of Physicians of Saxony, the mayors of the municipalities contactes during sampling, and the Public Health Department in Dresden for supporting the project.

## Author Contributions

**Conceptualization:** Lorenz Harst, Tina Haase, Falko Tesch, Markus Kösters, Jochen Schmitt.

**Data curation:** Lorenz Harst, Tina Haase, Falko Tesch, Lilly Rüthrich.

**Formal analysis:** Lorenz Harst, Tina Haase, Falko Tesch, Lilly Rüthrich.

**Investigation:** Lorenz Harst, Tina Haase, Falko Tesch, Lilly Rüthrich.

**Methodology:** Lorenz Harst, Tina Haase, Falko Tesch.

**Project administration:** Lorenz Harst, Tina Haase.

**Resources:** Markus Kösters, Jochen Schmitt.

**Supervision:** Markus Kösters, Jochen Schmitt.

**Visualization:** Lorenz Harst, Tina Haase, Falko Tesch.

**Writing – original draft:** Lorenz Harst, Tina Haase.

**Writing – review & editing:** Lorenz Harst, Tina Haase, Falko Tesch, Lilly Rüthrich, Markus Kösters, Jochen Schmitt.

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
