## [Decision Letter · Decision Letter 0]

31 Jul 2024

PONE-D-24-06039Rationale and study protocol of a regional health panel in Saxony, Germany (GEPASA)PLOS ONE

Dear Dr. Harst,

Thank you for submitting your manuscript to PLOS ONE. After careful consideration, we feel that it has merit but does not fully meet PLOS ONE’s publication criteria as it currently stands. Therefore, we invite you to submit a revised version of the manuscript that addresses the points raised during the review process. Please submit your revised manuscript by Sep 14 2024 11:59PM. If you will need more time than this to complete your revisions, please reply to this message or contact the journal office at plosone@plos.org. Please include the following items when submitting your revised manuscript:A rebuttal letter that responds to each point raised by the academic editor and reviewer(s). You should upload this letter as a separate file labeled 'Response to Reviewers'.A marked-up copy of your manuscript that highlights changes made to the original version. You should upload this as a separate file labeled 'Revised Manuscript with Track Changes'.An unmarked version of your revised paper without tracked changes. You should upload this as a separate file labeled 'Manuscript'.

We look forward to receiving your revised manuscript.

Kind regards,

Tai-Heng Chen, M.D., Ph.D.

Academic Editor

PLOS ONE

“I have read the journal's policy and the authors of this manuscript have the following competing interests: JS reports institutional grants for investigator-initiated research from the German GBA, the BMG, BMBF, EU, Federal State of Saxony, Novartis, Sanofi, ALK, and Pfizer. He also participated in advisory board meetings as a paid consultant for Sanofi, Lilly, and ALK. JS serves the German Ministry of Health as a member of the Sachverständigenrat Gesundheit und Pflege. MK reports institutional grants for investigator-initiated research from the German GBA, the EU, BMBF and the DFG. LH reports fees from the Volkswagen Foundation and from Thieme. LR and TH report fees from the aQua-Institut. These reports are unrelated to this study.”

Reviewers' comments:

Reviewer's Responses to Questions

**Comments to the Author**

1. Does the manuscript provide a valid rationale for the proposed study, with clearly identified and justified research questions?

Reviewer #1: Yes

Reviewer #2: Yes

2. Is the protocol technically sound and planned in a manner that will lead to a meaningful outcome and allow testing the stated hypotheses?

Reviewer #1: Yes

Reviewer #2: Yes

3. Is the methodology feasible and described in sufficient detail to allow the work to be replicable?

Reviewer #1: No

Reviewer #2: Yes

4. Have the authors described where all data underlying the findings will be made available when the study is complete?

Reviewer #1: Yes

Reviewer #2: Yes

5. Is the manuscript presented in an intelligible fashion and written in standard English?

Reviewer #1: Yes

Reviewer #2: Yes

6. Review Comments to the Author

You may also provide optional suggestions and comments to authors that they might find helpful in planning their study.

Reviewer #1: Overall, the study protocol is well written, follows a concise structure, and the arguments and descriptions are easy to understand. The authors did a very solid piece of work that should be acknowleged. As far as I can evaluate this, the English language is sufficient. In summary, I only have some very minor concerns:

1) Authorship: I wonder, if 4 authors in responsible first or last authorship really contributed equally to this manuscript. For scientific fairness, please check on the official guidelines for authorship.

2) Abstract, spelling mistake: “issues releated to the perceived quality of health”

3) Rationale, sentence correction needed: “Panel surveys are longitudinal studies researching the same sample time and again over a 99 potentially open-ended time period.”

4) Methods: Though the dimensions and contents of the planned questionnaire are described, the actual questions, variables and answering scales remain unclear. The set-up of the factual questions and scales will be of great importance to allow for the planned analysis. Can the authors proved additional information on the questionnaire?

5) Methods, Sampling: Who sends the initial invitation letter? Are the relevant residents' registration offices paid for their services in the study? How will recruitment be organised in the event of under-recruitment even lower than 15%? Who will organise the initial mailing and return and how? Who enters the data?

6) Statistical plan, spelling mistake: “questions were fromulated”

Reviewer #2: Thank you for the opportunity to review the resubmitted manuscript “Rationale and study protocol of a regional health panel in Saxony, Germany (GEPASA)”

The manuscript is well written and gives a good overview on the planned study. Some minor aspects should be addressed:

• Death rate per 100.000 population? (line 97)

• Rephrase “Panel surveys are longitudinal studies researching the same sample time and again over a potentially open-ended time period.” (line 98)

• How are rural communities are sampled? This is not clear and not described in the text.

• Please state if standardized instrument are used to address the research questions (e.g. quality of life). Using standardized instruments enables to compare results to other surveys.

• What is the duration of the panel survey?

• Figure 2 should be translated in English

• As (perceived) quality of care is focused on, more information on different quality of care aspects should be stated, like structural quality, process quality and quality of outcomes. Further, perceived quality of care should be contrasted to actual (measured) quality of care. This should be shortly discussed.

7. PLOS authors have the option to publish the peer review history of their article (what does this mean?). If published, this will include your full peer review and any attached files.

Reviewer #1: No

Reviewer #2: No

---

## [Author Response · Author response to Decision Letter 0]

8 Aug 2024

Dear esteemed reviewers at PLOS ONE,

Thank you for your valuable comments to our manuscript entitled “Rationale and study protocol of a regional health panel in Saxony, Germany (GEPASA)”. We have adapted several parts of our manuscript, as described below.

 Reviewer #1

1 Overall, the study protocol is well written, follows a concise structure, and the arguments and descriptions are easy to understand. The authors did a very solid piece of work that should be acknowleged. As far as I can evaluate this, the English language is sufficient. R: Thank you for this positive overall evaluation of our manuscript which strengthens our belief in the relevance of our work.

2 I wonder, if 4 authors in responsible first or last authorship really contributed equally to this manuscript. For scientific fairness, please check on the official guidelines for authorship. R: Thank you for raising awareness to the issue of fairness in authorship. However, we assure you that authorship of our paper is distributed and labeled in a manner that reflects the multiple and valuable contributions made by all authors to both the manuscript and the GEPASA project as a whole.

3 Abstract, spelling mistake: “issues releated to the perceived quality of health” R: Thank you, we have corrected the mistake.

4 Rationale, sentence correction needed: “Panel surveys are longitudinal studies researching the same sample time and again over a 99 potentially open-ended time period.” R: Thank you, we have corrected the mistake.

5 Methods: Though the dimensions and contents of the planned questionnaire are described, the actual questions, variables and answering scales remain unclear. The set-up of the factual questions and scales will be of great importance to allow for the planned analysis. Can the authors proved additional information on the questionnaire? R: Thank you for raising this important point. We have gladly added information on the final questionnaire to the manuscript, which read as follows:

“The complete questionnaire, containing both the questions (n = 48) and the scales, is available as a supplementary file to this publication. Within this file, we also report on the origin of each question, thus demonstrating we only use standardized – not open-ended – questions and mostly rely on previously used and such valid as well as reliable questions. We also explain each question’s anticipated use in subsequent data analyses. Thus, we also reflect on the role each question plays in the overall purpose of our panel study.” (p. 11, ll. 166ff)

As stated in the section above, we also provide a complete version of the questionnaire as an annex, reporting also on origin and purpose of each question.

6 Methods, Sampling: 1. Who sends the initial invitation letter? 2. Are the relevant residents' registration offices paid for their services in the study? 3. How will recruitment be organised in the event of under-recruitment even lower than 15%? 4. Who will organise the initial mailing and return and how? 5. Who enters the data? R: Thank you for your detailed interest in our sampling and recruitment procedure. The following information has been added to our manuscript:

1. “Letters to the individual participants are addressed and sent by the Data Trust Office of Technische Universität (TU) Dresden (see below) in order to ensure separation of the participants’ personal information and their responses in the questionnaire.” (p. 14, ll. 209ff)

2. “Administrative fees for drawing the sample are anticipated for some of the registration offices.” (p. 13, ll. 198f)

3. “In case of a response rate lower than 15 %, advertisement in official publications of the major’s offices in the included communities will be placed, explaining the purpose of the panel and stressing the fact that participation is a means for citizens to be heard on such an important issue as health care planning.” (p. 15, ll. 224ff)

4. “All return letters are collected at the project office within the responsible institute, where questionnaire and consent forms are separated. The latter are collected and managed by the Data Trust Office.” (p. 15, ll. 214ff)

5. “The questionnaire will be presented in paper and will also be available as an online version. The printed questionnaire will be created using the software FormPro and the digital questionnaire will be created using the software REDCap. Participants choosing the latter option will find an individualized link to the online questionnaire on REDCap within the recruitment papers. A stamped return envelope is included in the initial contact letter for participants deciding to complete the questionnaire on paper. Data entry from printed questionnaires will be largely automated using FormPro software. The accuracy of the readings will be checked manually. ” (p. 14, ll. 207ff)

7 Statistical plan, spelling mistake: “questions were fromulated” R: Thank you, we have corrected the mistake.

Reviewer #2

8 The manuscript is well written and gives a good overview on the planned study. R: Thank you for this positive overall evaluation of our manuscript which strengthens our belief in the relevance of our work.

9 Death rate per 100.000 population? (line 97) R: Thank you for suggesting an alternative for reporting the death rate. We have now reported the rate for 2022 per 100,000 people:

“Additionally, Saxony had the 3rd largest death rate of all German federal states, independent of the cause of death (1.522 in 20211,480 deaths per 100,000 citizens in 2022).” (p. 7, l. 102)

10 Rephrase “Panel surveys are longitudinal studies researching the same sample time and again over a potentially open-ended time period.” (line 98) R: Thank you, we have corrected the mistake.

11 How are rural communities are sampled? This is not clear and not described in the text. R: Thank you for shining light on the important issue of rural sampling. We have elaborated on this issue further in the manuscript:

“Municipalities largely represent rural areas in Saxony and were drawn based on their postal code. In order to ensure a clear differentiation of the area of residence of the participants, municipalities with identical postal codes were not included in the sample.” (p. 12, ll. 191ff)

12 Please state if standardized instrument are used to address the research questions (e.g. quality of life). Using standardized instruments enables to compare results to other surveys. R: Thank you for raising this important point. We have gladly added information on the origin of our questions to the manuscript:

“The complete questionnaire, containing both the questions (n = 48) and the scales, is available as a supplementary file to this publication. Within this file, we also report on the origin of each question, thus demonstrating we only use standardized – not open-ended – questions and mostly rely on previously used and such valid as well as reliable questions.” (p. 11, ll. 166ff)

In the discussion, we have revisited the subject, stating once more the value of standardized instruments:

“Also, by mostly using previous used and such valid and reliable questions, the results of the panel surveys can be compared to at least national benchmarks.” (p. 18, ll. 276ff)

As stated in the section above, we also provide a complete version of the questionnaire as an annex, reporting also on origin of each question.

13 What is the duration of the panel survey? R: Our panel is planned as an open-ended project, which we now state in the manuscript on p. 15, l. 244f (section “Status and Timeline”).

14 Figure 2 should be translated in English R: Thank you for pointing this out. We have changed the figure to an English version.

15 As (perceived) quality of care is focused on, more information on different quality of care aspects should be stated, like structural quality, process quality and quality of outcomes. Further, perceived quality of care should be contrasted to actual (measured) quality of care. This should be shortly discussed. R: Thank you for reminding us to take a closer look on dimensions of quality. We have now done so in the introduction to our manuscript:

“…the population's perspective on the availability of doctors, hospitals or other health care facilities, on wait times or on travel distances to health care facilities is still not considered within the federal planning process. These issues, however, are dimensions of quality of health care, covering structural (e.g. availability, distances), process (e.g. waiting times) as well as outcome (experiences made during consultations with health care professionals) quality alike [12]. While some of these domains (availability, distances) can be measured using, e.g., routine data from health care, others can only be assessed by asking (potential) patients directly about them [13].” (p. 6, ll. 83ff)

In the cited section, we also touch upon the issue of objective and subjective measures of quality. We raise this issue once more in the discussion, stating the importance of studying quality of health care using both routine health care data and subjective measures of perceived quality of health care:

“Furthermore, data gained on perceived quality of health care can be used complementary to routinely collected data on health care and such contribute to a more holistic health care monitoring [25].” (p. 17 ll. 275ff)

---

## [Decision Letter · Decision Letter 1]

5 Sep 2024

Rationale and study protocol of a regional health panel in Saxony, Germany (GEPASA)

PONE-D-24-06039R1

Dear Dr. Harst,

We’re pleased to inform you that your manuscript has been judged scientifically suitable for publication and will be formally accepted for publication once it meets all outstanding technical requirements.

Kind regards,

Tai-Heng Chen, M.D., Ph.D.

Academic Editor

PLOS ONE

Reviewers' comments:

Reviewer's Responses to Questions

**Comments to the Author**

1. Does the manuscript provide a valid rationale for the proposed study, with clearly identified and justified research questions?

Reviewer #1: Yes

Reviewer #3: Yes

2. Is the protocol technically sound and planned in a manner that will lead to a meaningful outcome and allow testing the stated hypotheses?

Reviewer #1: Yes

Reviewer #3: Yes

3. Is the methodology feasible and described in sufficient detail to allow the work to be replicable?

Reviewer #1: Yes

Reviewer #3: Yes

4. Have the authors described where all data underlying the findings will be made available when the study is complete?

Reviewer #1: Yes

Reviewer #3: Yes

5. Is the manuscript presented in an intelligible fashion and written in standard English?

Reviewer #1: Yes

Reviewer #3: Yes

6. Review Comments to the Author

You may also provide optional suggestions and comments to authors that they might find helpful in planning their study.

Reviewer #1: Considering my own review, the authors carefully took my comments into account and corrected the manuscript accordingly. For my part, no further changes are necessary.

Reviewer #3: The authors aimed to conduct a longitudinal panel study of the perceived health status and perceived quality of health care of the Saxon citizens as a basis for the systematic derivation of health care goals/measures and target group-specific, regionally suitable prevention measures The citizens' perspectives on health care are central to the assessment of the health care situation and to regional developmenti in Saxony, Germany, which is very interesting for potential readers in such field and well-written. The responses to the the reviewers were wonderful and I have no more questions about it.

7. PLOS authors have the option to publish the peer review history of their article (what does this mean?). If published, this will include your full peer review and any attached files.

Reviewer #1: No

Reviewer #3: **Yes: **HE Zhiqing

---

## [Editor Report · Acceptance letter]

17 Sep 2024

PONE-D-24-06039R1 

PLOS ONE

Dear Dr. Harst, 

I'm pleased to inform you that your manuscript has been deemed suitable for publication in PLOS ONE. Congratulations! Your manuscript is now being handed over to our production team.

Kind regards, 

on behalf of

Dr. Tai-Heng Chen 

Academic Editor

PLOS ONE